# A generative model of electrophysiological brain responses to stimulation

Diego Vidaurre[1,2]*

[1]Center for Functionally Integrative Neuroscience, Department of Clinical Medicine, Aarhus University, Aarhus, Denmark; [2]Department of Psychiatry, Oxford University, Oxford, United Kingdom

*For correspondence:
dvidaurre@cfin.au.dk

Competing interest: The author declares that no competing interests exist.

**Abstract** Each brain response to a stimulus is, to a large extent, unique. However this variability, our perceptual experience feels stable. Standard decoding models, which utilise information across several areas to tap into stimuli representation and processing, are fundamentally based on averages. Therefore, they can focus precisely on the features that are most stable across stimulus presentations. But which are these features exactly is difficult to address in the absence of a generative model of the signal. Here, I introduce *genephys*, a generative model of brain responses to stimulation publicly available as a Python package that, when confronted with a decoding algorithm, can reproduce the structured patterns of decoding accuracy that we observe in real data. Using this approach, I characterise how these patterns may be brought about by the different aspects of the signal, which in turn may translate into distinct putative neural mechanisms. In particular, the model shows that the features in the data that support successful decoding—and, therefore, likely reflect stable mechanisms of stimulus representation—have an oscillatory component that spans multiple channels, frequencies, and latencies of response; and an additive, slower response with a specific (cross-frequency) relation to the phase of the oscillatory component. At the individual trial level, still, responses are found to be highly variable, which can be due to various factors including phase noise and probabilistic activations.

## eLife assessment

This study presents a **valuable** finding on developing a state-of-the-art generative model of brain electrophysiological signals to explain temporal decoding matrices widely used in cognitive neuroscience. The evidence supporting the authors' claims is **convincing**. The results will be strengthened by providing more clear mappings between neurobiological mechanisms and signal generators in the model. The work will be of interest to cognitive neuroscientists using electrophysiological recordings.

## Introduction

There are virtually infinite manners by which a constant stimulus can impinge into the sensorium of an animal. For instance, our noses have many receptors that can sense a given odorant molecule, but only a small subset of those are excited each time the odour is perceived (*Axel, 1995*). Similarly, photons hit and excite photoreceptors in the retina randomly and sparsely for a given presented image (*Dowling, 1987*). An important aspect of sensation and perception is that never the exact same receptors are excited every time we perceive and, still, our perceptual experiences feel quite stable. So the brain must have a way to transit from lack of invariance at the microscopic sensory

level towards invariance at the macroscopic level, which ultimately supports the invariant aspects of conscious perception and behaviour. However, we observe significant variability in the brain responses at the trial level (*Stein et al., 2005*; *McIntosh et al., 2008*; *Garrett et al., 2013*), including at the earliest layers of the perceptual hierarchy (*Croner et al., 1993*; *Freeman, 1978*)—that is, each perceptual experience is associated with a unique neural trajectory that does not repeat. How the gap between stability in subjective perception and the changing nature of brain responses is bridged is an important question in neuroscience. Here, I investigate the distributed aspects of brain activity that are most stable across experimental repetitions, and are therefore most likely to relate to stable perceptual experiences.

Decoding analysis uses multivariate machine-learning algorithms to predict the identity of the observed stimulus from brain data (*Haxby et al., 2014*; *Stokes et al., 2015*). This way, per time point, it estimates a function of the data that maximally discriminate between conditions, as well as a temporal measure of accuracy that reflects how much information the data convey about the stimuli per time point. The assessment, via decoding accuracy, of how the discriminative space changes throughout the trial offers a view of the properties of stimuli representation and processing. However, it is not straightforward to know what specific aspects of the signal cause the patterns of decoding accuracy that we observe in perceptual experiments. Without this capacity, it is hard to link these patterns to actual neural mechanisms.

To gain insight into what aspects of the signal underpin decoding accuracy, and therefore the most stable aspects of stimulus processing, I introduce a generative model of multichannel electrophysiological activity (e.g. electroencephalography[EEG] or magnetoencephalography [MEG]) that, under no stimulation, exhibits chaotic phasic and amplitude fluctuations; and that, when stimulated, responds by manipulating certain aspects of the data, such as ongoing phase or signal amplitude, in a stimulus-specific manner. Specifically, in every trial, each channel may or may not respond to stimulation, according to a certain probability. In the model, when a channel responds, it can do it in different ways: (1) by phase resetting the ongoing oscillation to a given target phase and then entraining to a given frequency, (2) by an additive oscillatory response independent of the ongoing oscillation, (3) by modulating the amplitude of the stimulus-relevant oscillations, or (4) by an additive non-oscillatory (slower) response. This (not exhaustive) list of effects was considered given previous literature (*Shah et al., 2004*; *Mazaheri and Jensen, 2006*; *Makeig et al., 2002*; *Vidaurre et al., 2021*), and each effect may be underpinned by distinct neural mechanisms. For example, it is not completely clear the extent to which stimulus processing is sustained by oscillations, and disentangling these effects can help resolving this question. I named this model *genephys* by *gen*erative model of *e*mpirical electro-*phys*iological signals. Genephys is empirical in the sense that it purely accounts for features in the signal that are observable, without making assumptions about the underlying neurobiological causes; that is, it can generate signals that, with the right parametrisation, can share empirical properties with real data. In particular, when confronted with a decoding algorithm, the data generated by this model can show patterns of decoding accuracy with similar characteristics to what we observe in real data.

Given the effects that we observe in the stimulus processing literature, and using an example of visual perception as a reference, I observed that two different mechanisms can produce realistic decoding results as we see in real perception: either phase resetting to a stimulus-specific phase followed by frequency entrainment, or an additive oscillation (unrelated to the ongoing oscillations) with a stimulus-specific phase. Either way, a cross-frequency coupling effect is also necessary, where an additive slower response holds a specific phasic relation with the oscillatory (faster) response. Furthermore, the stimulus-related oscillation needs to span multiple channels, and have a diversity of frequencies and latencies of response across channels. Other experimental paradigms, including motor tasks and decision making, can be investigated with g*enephys*, which is publicly available as a Python package in PyPI http://github.com/vidaurre/genephys; http://genephys-doc.readthedocs.io/en/latest/.

## Methods
### A generative model of empirical electrophysiological signals: *genephys*

While the system is unperturbed, *genephys* is based on sampling spontaneously varying instantaneous frequency and amplitude (i.e. square root of power) time series, $f^{\mathrm{rest}}$ and $a^{\mathrm{rest}}$, respectively,

that are analytically combined to form the sampled signal $x$. Amplitude and frequency are allowed to oscillate within ranges $r^f$ and $r^a$. Instantaneous frequency here refers to angular frequency, from which we can obtain the ordinary frequency in Hertz as $\frac{F}{2\pi} f^{\text{rest}}$, where $F$ is the sampling frequency of the signal. Both $f^{\text{rest}}$ and $a^{\text{rest}}$ are sampled separately from autoregressive processes of order one, endowing them with chaotic, non-oscillatory dynamics. Specifically, given autoregressive parameters $b^f, b^a < 1$, and Gaussian-noise variables $e_{f_t}, e_{a_t}$, I generate $f^{\text{rest}}$ and $a^{\text{rest}}$ for a given channel as

$$f_0^{\text{rest}} = e_{f_0}$$
$$f_t^{\text{rest}} = b_f f_{t-1}^{\text{rest}} + e_{f_t} \text{ for t} > 0$$
$$a_0^{\text{rest}} = e_{a_0}$$
$$a_t^{\text{rest}} = b_a a_{t-1}^{\text{rest}} + e_{a_t} \text{ for t} > 0$$

Without stimulation, a phase time series $\varphi^{\text{rest}}$ is then built as

$$\varphi_0^{\text{rest}} = 0$$
$$\varphi_t^{\text{rest}} = \varphi_{t-1}^{\text{rest}} + f_t^{\text{rest}}$$

Then, given some Gaussian-distributed measurement noise $\epsilon_t$ with standard deviation $\sigma_\epsilon$, I build $x$ (in absence of stimulation) as

$$x_t^{\text{rest}} = a_t^{\text{rest}} \sin \varphi_t^{\text{rest}} + \epsilon_t$$

This process is done separately per channel and per trial. Note that, under no stimulation, the channel time series are (asymptotically) uncorrelated. We can think of them as dipoles in brain space. We can induce correlations for instance by projecting these time series onto a higher-dimensional space, which we can consider to be in sensor space, or by using correlated noise. Altogether, this generates chaotic oscillatory data relatively akin to real data.

When a stimulus $k$ is presented at time point $\tau$ of the trial, a perturbation is introduced into the system on the $p$ channels that are stimulus relevant (which can be a subset of the total number of channels). When a channel is not relevant, it does not respond; when it is relevant, it responds to the stimulus with a given probability:

Prob (channel $j$ responds | channel $j$ is relevant) = $\theta_j$
Prob (channel $j$ responds | channel $j$ is not relevant) = 0

where $\theta_j$ is a hyperparameter representing a channel-specific probability. In simpler words, relevant channels might respond in some trials, but not in others. In all the simulations, I set all channel probabilities to a single value, $\theta_j = \theta$, but other configurations are possible.

Given some stimulus presentation at time point $\tau$, a channel may respond:

- By phase resetting to a condition-specific target phase, and then, when the target phase is reached, by frequency entraining to a given target frequency.
- By adding a transient (damped), oscillatory response with a condition-specific phase, amplitude, and frequency.
- By increasing the (absolute) amplitude of the ongoing oscillation following stimulus presentation (with no phase effect).
- By adding a transient, non-oscillatory response with a condition-specific amplitude.

The timing of the effect is controlled by a response function. I use an double logarithmic response function, asymmetric around the time of maximum effect $t_{\text{max}}$. For the left side (i.e. before $t_{\text{max}}$), this function is parametrised by: $\delta_1$, reflecting the latency of the response in number of time points; and $\delta_2$, reflecting how many time points it takes the logarithmic function to go from zero to its maximum before it changes to the right-side logarithmic function. Therefore, $t_{\text{max}} = \delta_1 + \delta_2 + \tau$. I introduce some noise in $\delta_1$ per trial to make the timing of the response to vary, as per $\delta_1 \sim \text{Uniform}\left(0, \sigma_{\delta_1}\right)$, where $\sigma_{\delta_1}$ is a model hyperparameter. This latency noise could be fixed for all channels (absolute stochastic latency) or has a value per channel (relative stochastic latency; *Vidaurre et al., 2021*). It is also possible to use different response function parametrisations per channel. For example, we can

induce a diversity of latencies for the different channels by using different values of $\delta_1$ per channel, so that some channels (e.g. those more closely related to primary sensory areas) respond earlier than others (e.g. those related to associative areas). Once the activation reaches its maximum at $t_{\max}$, the decay is also parametrised by a logarithmic function with a parameter $\delta_3$, reflecting how many time points it takes the logarithmic function to go from its maximum (at $t_{\max}$) to zero. A different response function can be used for each type of effect, which can be combined. The next subsection provides a full mathematical specification of the response function.

With respect to the phase reset and frequency entrainment effect, the phase reset occurs before $t_{\max}$, when the target phase is reached. Given condition or stimulus $k$, for each trial and channel,

$$\varphi_t = \varphi_{t-1} + \left(1 - g_t\right) f_t^{\text{rest}} + g_t \nabla_t$$

with

$$\nabla_t = \begin{cases} \varphi^k - \varphi_t, \text{if} \varphi^k - \varphi_t \in \left[-\pi, \pi\right] \\ \varphi^k - \varphi_t - 2\pi, \text{if} \varphi^k - \varphi_t > \pi \\ \varphi^k - \varphi_t + 2\pi, \text{if} \varphi^k - \varphi_t < \pi \end{cases}$$

where, for each trial, $\varphi^k$ is randomly sampled from a von Mises distribution with mean equal the target phase $\bar\varphi^k$ and standard deviation $\sigma_\varphi$; $\nabla_t$ is the polar gradient between the target phase and the ongoing phase $\varphi_{t-1}$; and $g_t$ is the value taken by the response function at time point $t$. After $t_{\max}$, phase resetting is over, and the phase entrainment period starts,

$$\varphi_t = \varphi_{t-1} + g_t f^k + \left(1 - g_t\right) f_t^{\text{rest}}$$

The system then entrains to a possibly stimulus-specific, possibly channel-specific, target frequency $f^k$, with a strength that logarithmically decreases as $t$ moves away from $t_{\max}$.

With respect to the additive oscillatory response, we consider a sinusoidal oscillator, which is damped by the action of the response function $g_t$:

$$y_t = g_t \alpha^k \sin\left(\omega^k t + \gamma^k\right)$$

Here, $\alpha^k$ reflects the amplitude of the additive oscillation, $\omega^k$ its frequency, and $\gamma^k$ its phase.

For the amplitude modulation, I apply a multiplying factor to the ongoing amplitude time series:

$$a_t = \left(1 + g_t m^k\right) a_t^{\text{rest}}$$

where $m^k$ is a proportional increment; for example, $m^k = 0.1$ would produce an increment of 10% in amplitude at $t_{\max}$.

With respect to the additive non-oscillatory response, we have:

$$z_t = g_t s^k$$

where $s^k$ is sampled from a Gaussian distribution, $\mathcal{N}\left(\bar z^k, \sigma_z\right)$, where $\bar z^k$ is stimulus specific.

Given these elements, the signal is built as

$$x_t = a_t \sin \varphi_t + z_t + y_t + \epsilon_t$$

This model can be trivially extended to have correlated noise $\epsilon_t$ across channels.

*Figure 1* shows two examples of how the effect of stimulation looks for one trial and one channel. The left panel corresponds to a phase resetting plus frequency entrainment effect, and the middle panel corresponds to an additive oscillation; both are accompanied by an additive non-oscillatory response. Here, the sampled signal $x$ is shown in blue on top, and the phase $\varphi$, frequency $f$, amplitude $a$, additive non-oscillatory response $z$, and additive oscillatory response $y$ are shown in red underneath. For comparison, the right panel shows real MEG data from a passive visual experiment where a number of images are shown to the subjects at a rate of one image per second (*Cichy et al.,*

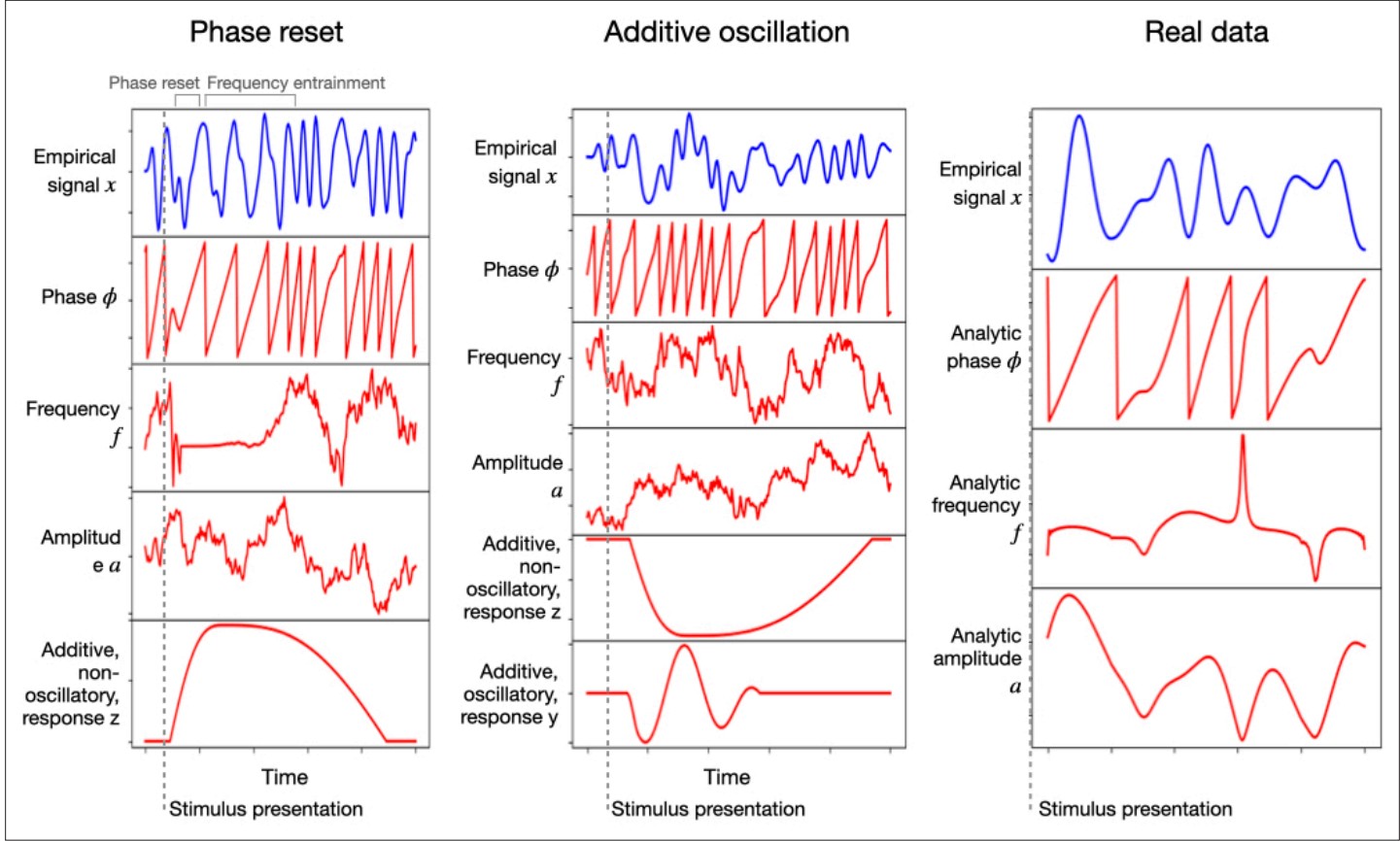

**Figure 1.** Left and middle: single-trial example of the generated signal (in blue) and its constitutive components (in red): instantaneous phase, frequency, and amplitude, as well as an additive non-oscillatory and oscillatory response components; the left panel reflects a phase resetting plus frequency entrainment effect, while the middle panel corresponds to an additive oscillatory response. Right: real (filtered) magnetoencephalography data collected during passive stimuli viewing; the red curves are the analytically computed phase, frequency, and amplitude (via the Hilbert transform).

*2016*); the measured (filtered) signal is shown in blue, while the red curves correspond to analytical phase, frequency, and amplitude (computed via the Hilbert transform on the filtered signal). *Table 1* summarises all the hyperparameters that configure the model.

## A full mathematical description of the response function

Assuming the stimulus was presented at time point $\tau$ within a given trial, and that both $t$ and $\tau$ are expressed in number of time points, the response function is asymmetric around $t_{\max}$, when the function takes the value 1.0 from both sides. In the experiments above, both left and right parts are logarithmic. Mathematically,

$$g_t = G\left(t;\tau\right) = 0.0, \text{if } t < \delta_1 + \tau$$

$$g_t = G\left(t;\tau\right) = -\log\left(1 + \left(\frac{\delta_1 + \delta_2 + \tau - t - 1}{T_1}\right)^{\varsigma_1}\right) + C_1, \text{if } t \in \left[\delta_1 + \tau, \delta_1 + \delta_2 + \tau\right]$$

$$g_t = G\left(t;\tau\right) = -\log\left(1 + \left(\frac{t - \tau - \delta_1 - \delta_2 - 1}{T_2}\right)^{\varsigma_2}\right) + C_2,$$

$$\text{if } t \in \left[\delta_1 + \delta_2 + \tau, \delta_1 + \delta_2 + \delta_3 + \tau\right]$$

$$g_t = G\left(t;\tau\right) = 0.0, \text{if } t > \tau + \delta_1 + \delta_2 + \tau + 1 + T_2$$

where
- $\delta_1$ reflects the latency of the response in number of time points,
- $\delta_2$ is how many time points it takes the left logarithmic function to go from 0.0 to 1.0,
- $\delta_3$ is how many time points it takes the right logarithmic function to go from 1.0 to 0.0,

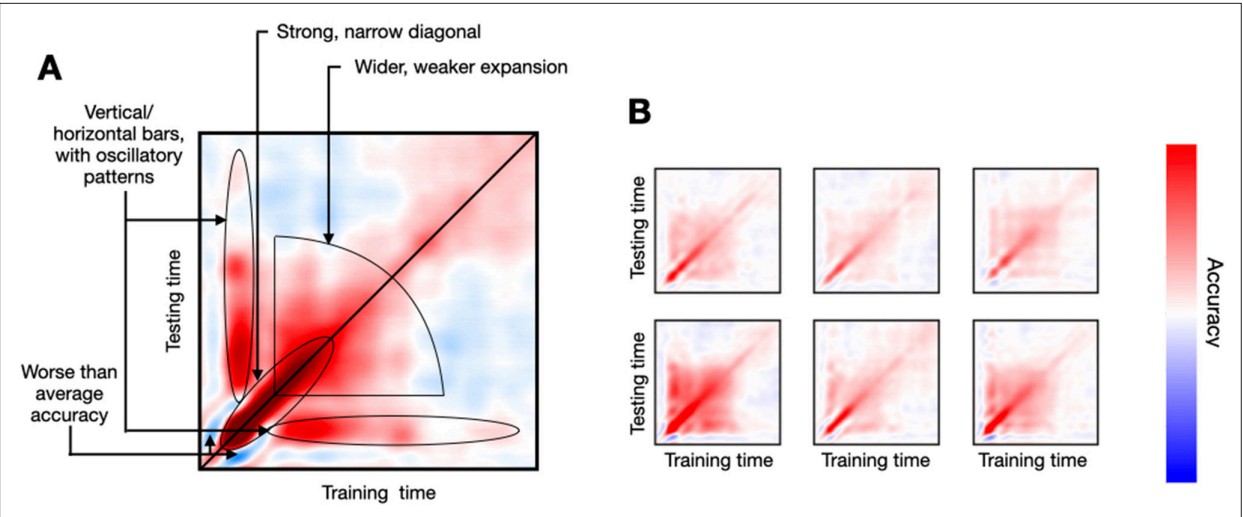

**Figure 2.** Empirical decoding results from (real) data, where subjects underwent a passive viewing experiment. (**A**) One example subject's temporal generalisation matrix (TGM), where the different characteristic features have been highlighted. (**B**) TGMs from six example subjects.

- $t_{\max} = \delta_1 + \delta_2 + \tau$ is the time point of maximum response (i.e. the changing point between the two functions),

**Table 1.** *genephys* configuration hyperparameters.

| Parameter | Math. notation |
|---|---|
| Frequency autoregressive parameter | $b_f$ |
| Amplitude autoregressive parameter | $b_a$ |
| Frequency range | $r_f$ |
| Amplitude range | $r_a$ |
| Measurement noise standard deviation | $\sigma_\epsilon$ |
| Channel activation probability | $\theta$ |
| Number of relevant channels | $p$ |
| Response function—latency | $\delta_1$ |
| Response function—rise slope | $\delta_2$ |
| Response function—fall slope | |
| Target phase, mean | $\overline{\varphi^k}$ |
| Target phase, standard deviation | $\sigma_\varphi$ |
| Additive oscillatory response, phase | $\gamma^k$ |
| Additive oscillatory response, amplitude | $\alpha^k$ |
| Additive oscillatory response, frequency | $\omega^k$ |
| Amplitude modulation | $m^k$ |
| Additive non-oscillatory response, mean | $\overline{z^k}$ |
| Additive non-oscillatory response, standard deviation | $\sigma_z$ |

- $C_1$, $C_2$, $T_1$, and $T_2$ are normalisation constants chosen such that the logarithmic functions are bounded between 0.0 and 1.0, and $G(t_{\max}; \tau) = 1.0$ from both sides,
- $\varsigma_1$ and $\varsigma_2$ determine the shape of the logarithmic functions (here chosen to 2 and 4, respectively).

Note that, thanks to the normalisation constants, both the left and the right sides of the response function take values between 0.0 and 1.0. There is also the possibility of using an exponential function—which is faster to decay—in either side, but I did not use it in the experiments. For example, in the left part, this would take the form:

$$\frac{e^{-\dfrac{0.5\left(\delta_1 + \delta_2 + \tau - t\right)^2}{C_3}}}{C_4},$$

where $C_3$ and $C_4$ are normalisation constants.

## Decoding accuracy to characterise structured invariance

One possible approach to characterise the stable aspects of brain responses to stimulation is the analysis of average evoked responses potentials or fields (ERP/F) (*Dawson, 1954*; *Luck and Kappenman, 2011*; *Pfurtscheller and Lopes da Silva, 1999*). However, this approach is limited because perceptual experiences emerge from activity patterns across multiple brain areas acting in a distributed fashion, whereas ERP/Fs are

separately evaluated channel by channel. Also, ERP/F analyses are not concerned with what aspects of the signal carry specific information about the stimulus—that is they are not predictive of the stimulus (**Kragel et al., 2018**).

I instead focus on decoding, which finds, in a multivariate fashion, patterns of differential activity between conditions across channels and throughout time (**Grootswagers et al., 2017**). I use linear discriminant analysis, which estimates a projection or subspace in the data that maximally discriminate between conditions. Throughout the trial, this set of projections reflects information about the dynamics of stimulus processing.

As the read-out of decoding analysis, I use the temporal generalisation matrix (TGM), a $T \times T$ matrix of decoding accuracies (where $T$ is the number of time points in the trial), such that one decoding model is trained per time point and tested on every time point of the trial using cross-validation (**King and Dehaene, 2014**). The diagonal of the TGM reflects how well we can decode information time point by time point, while the off-diagonal shows how well decoding models generalise to time points different from those where they were trained. As illustrated in **Figure 2A**, where I show a real-data TGM from a subject performing a simple visual task (**Cichy et al., 2016**), the TGM exhibits some characteristic features that we often see throughout the literature. First, there is a strong diagonal band that is relatively narrow early in the trial, often surrounded by areas of worse-than-baseline accuracy. Then, the accuracy on the diagonal becomes progressively wider and weaker, expanding until relatively late in the trial. Also, there are bands of higher-than-baseline accuracy stemming vertically and horizontally from the time points of maximum accuracy (after a brief period of depression), which often show oscillatory behaviour throughout the band. **Figure 2B** presents TGMs from six subjects, where these features can also be appreciated yet with substantial variability across subjects. Below, I explore *genephys*' configuration space in relation to its ability to produce patterns of decoding accuracy similar to these that we observe in real data.

## Results

I confronted *genephys* to classification-based decoding analysis to characterise how brain activity carries stimulus-specific information. In each simulation, I generated 10 data sets per combination of parameters, each with $N = 250$ trials and $T = 250$ time points per trial (1 s for a sampling frequency of 250 Hz). The number of channels is 32. Only one endogenous oscillation was sampled, with (angular) frequencies spontaneously ranging between 0.01 and $0.25\pi$ (0.4–10 Hz). I computed a TGM per run and took the average across runs.

For reference, I considered MEG data recorded while participants viewed object images across 118 categories, as presented in **Cichy et al., 2016**. Each image category was presented 30 times. Presentation occurred during the first 500 ms of the trial, and trials were 1 s long, sampled at 250 Hz. The multi-channel sensor-space data, epoched around the presentation of each visual stimulus, can be used to train a decoder to predict which visual image is being presented (**Cichy et al., 2016**; **Vidaurre et al., 2021**). Here, I decoded whether the image corresponded to an animate category (a dog) or inanimate (a pencil). **Figure 2A** shows a TGM for an example subject, where some archetypal characteristics are highlighted. In the experiments below, specifically, I focus on the strong narrow diagonal at the beginning of the trial, the broadening of accuracy later in the trial, and the vertical/horizontal bars of higher-than-chance accuracy. Importantly, given the variability observed across subjects (as seen in **Figure 2B**, which shows TGMs for six subjects), this example is only meant as a reference; therefore I did not optimise the model hyperparameters to this TGM (except in the last subsection), or showed any quantitative metric of similarity.

### Oscillatory components underlying the observed decoding patterns

In real data, we often see oscillatory patterns in the TGM, indicating that the subspace of brain activity that carries information about the stimulus must have oscillatory elements. At least two distinct mechanisms may be behind this phenomenon: first, an ongoing oscillation might reset its phase and then entrain to a given frequency in a stimulus-specific fashion; second, a stimulus-specific oscillatory response might by added to the signal after stimulus presentation on top of the existing ongoing oscillation. Essentially, the difference between the two is that, in the phase-resetting case, the ongoing oscillations are altered; while in the other case the additive oscillation coexists with the

ongoing oscillations. Next, I use *genephys* together with decoding analysis to compare between these two alternatives, showing that both can produce decoding patterns similar to what we observe empirically in real experiments.

In the simulations, all 32 channels convey information but with a relatively low activation probability ($\theta = 1/6$). For phase resetting and frequency entrainment, I considered a diverse range of entrainment frequencies (between 0.1 and 0.2 in angular frequency) and latencies of response ($\delta_1 = 0 - 160$ms) across channels. For the additive oscillatory response, I considered a similar range of frequencies and latencies of response across channels. (I will show below that channel stochasticity and frequency diversity are both important to produce realistic decoding patterns.) The difference between the two fictitious stimuli lied in their different target frequencies (i.e. $\bar{\varphi}^k$ for phase resetting and $\gamma^k$ for the additive oscillation; see specification of the model in Methods). I also included an additive non-oscillatory response with a stimulus-specific amplitude, which (as I will also show later) is important to produce realistic decoding results. I did not optimise the parameters of the model to reflect the fine details

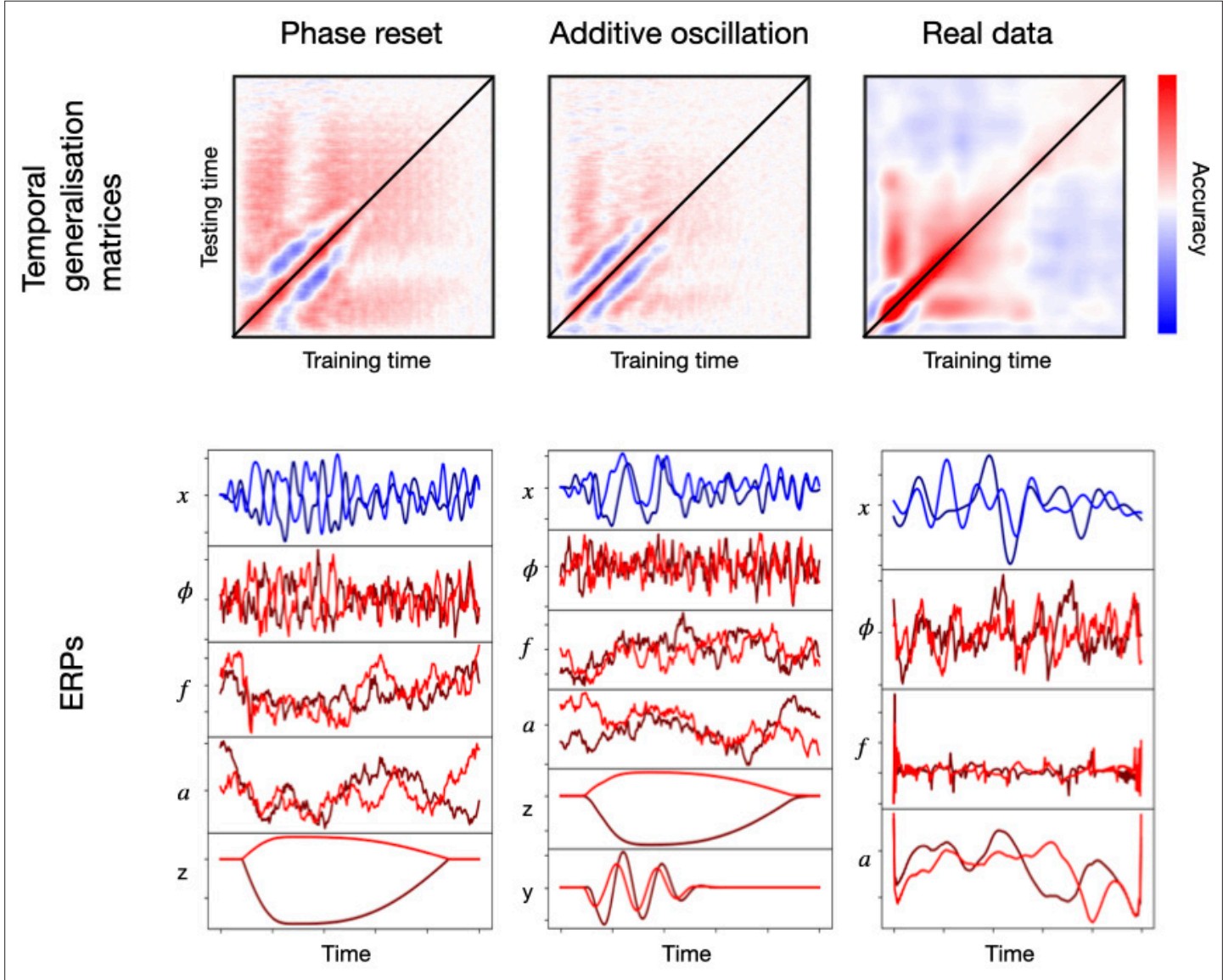

**Figure 3.** Examples of two configurations based on phase resetting and frequency entrainment (left) and additive oscillatory responses (middle), shown together with results obtained from real data (right). The top panels show temporal generalisation matrices (TGMs) from standard decoding analysis. The bottom panels show average evoked responses (ERP/F) for the sampled signal (blue) and its components (red); the two stimuli are represented with different tonalities of blue or red.

from real-data TGMs, since TGMs vary across subjects (see *Figure 2B*) and experimental paradigms (*King and Dehaene, 2014*).

*Figure 3* shows TGMs (top) together with one-channel ERPs (where each stimulus is represented by a different tonality of blue or red; bottom) for the sampled signal $x$ and its various constitutive elements: ongoing phase $\varphi$, frequency $f$, amplitude $a$, additive non-oscillatory response $z$, and, when applicable, an additive oscillatory response $y$. The left panels show results for phase reset plus frequency entrainment, where we can see an effect on the ongoing phase. The middle panels show results for the additive oscillation; here, there is no effect on the ongoing phase, and, instead, the additive oscillatory response $y$ is shown at the bottom. The right panels show an example from real data, where phase, frequency, and amplitude were computed analytically using the Hilbert transform on the filtered data.

Although the exact details differ from the right panels of the figure, both types of effects produce patterns reproducing the characteristic signatures of real data. These include the strong diagonal, the vertical/horizontal bands of high generalisation accuracy, and the broadening of accuracy in later stages of the trial. Note that phase resetting plus frequency entrainment is, everything else held constant, a stronger effect than the additive oscillatory response. This is because, for the latter, the ongoing oscillations (here, non-stimulus specific) can interfere with the phase of the additive response, impeding cross-trial phase locking throughout the trial. For phase resetting, on the other hand, the ongoing oscillation *is* the effect and there is no interference. In this particular example, anyway, the range for the additive oscillation (0.1–0.2) was much narrower than that of the ongoing oscillation (0.01–0.25$\pi$), making the interference more unlikely; that is, the ongoing oscillation phase is approximately averaged out, and treated as noise by the decoding algorithm.

## The distribution of stimulus-specific information spans multiple channels and is stochastic

Next, I use *genephys* to show that the stimulus-specific information spans a large number of channels, and do so stochastically. I focus on the additive oscillatory response effect, which, as shown in the previous section, can produce comparable results to phase resetting plus frequency entrainment.

I varied the number of relevant channels $p$ as well as the probability of activation $\theta$ for the relevant channels, so that the subset of channels that activate is different for every trial. *Figure 4* shows the TGMs for various combinations of $p$ and $\theta$, from a configuration where there are few relevant channels that always respond ($p = 1$, $\theta = 1$) to another where there are many relevant channels that only respond sparsely ($p = 32$, $\theta = 1/8$). As previously, channels have diverse frequencies and latencies of response. An additive non-oscillatory response is also included as an effect.

Contrary to empirical TGMs (which have a relatively stylised diagonal), having only a few relevant channels (first three panels of *Figure 4*) produces unrealistically geometric patterns. This indicates that, in real data, the subspace of the data that contains information about the stimulus needs to be multi-dimensional; that is, that the amount of relevant channels must be relatively high (as in the fourth panel of *Figure 4*). However, the contribution of each channel must entail some sort of noise or instability (in this case expressed by having probabilistic activations), or else the decoding accuracy becomes unrealistically perfect. In summary, the subspace of brain activity that carries out information about the condition is highly stochastic at the single-trial level.

## The oscillatory effect spans multiple frequencies and latencies

In the previous sections, I showed that a noisy, additive oscillation effect (or, alternatively, a phase reset plus frequency entrainment effect) across multiple channels can generate decoding patterns as we see in real data. Next, I demonstrate that the effect must span a diversity of frequencies across channels, as well as a diversity of latencies of response. In real data, frequency diversity can be expressed as, for example, a gradient of frequencies from primary towards more associative areas; while latency diversity could reflect phenomena such that primary areas responding earlier than associative areas.

For frequency diversity, each channel is endowed with a different effect-related frequency, such that frequencies are not multiples of each other (specifically, they have different values of $\omega^k$ between 0.1 and 0.2 in angular frequency; see Methods). For latency diversity, channels do not respond simultaneously but with different values of $\delta_1$ between 0 and 120 ms. *Figure 5* shows a two-by-two design. On the left column, I set *genephys* to have a uniform frequency of response (i.e. all channels have

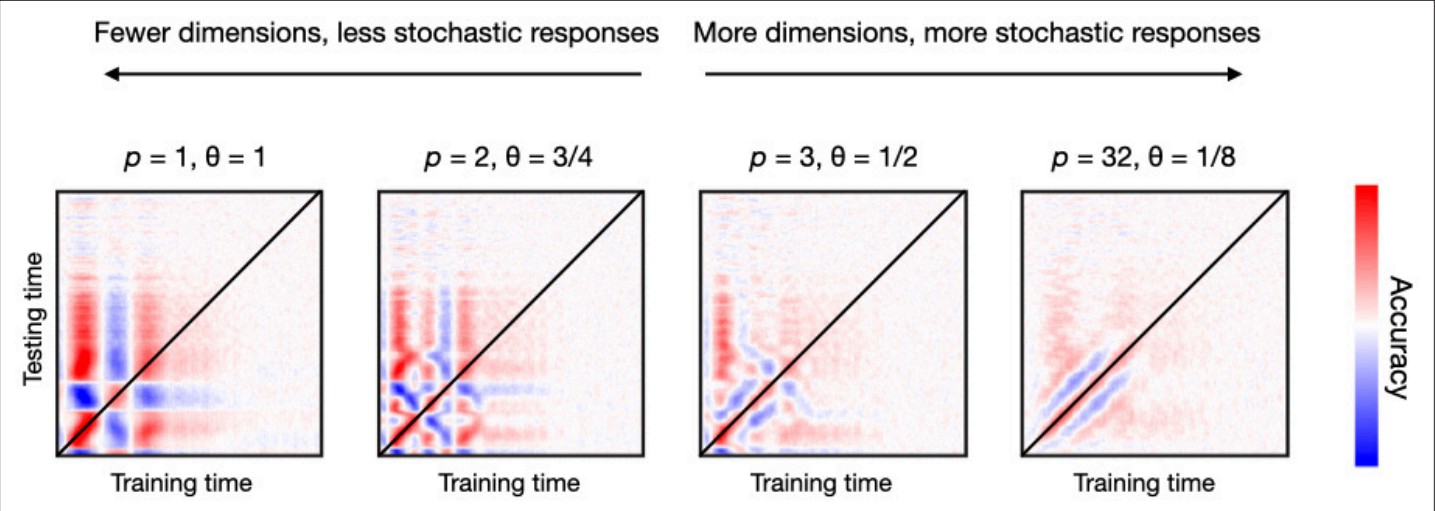

**Figure 4.** Having more dimensions (channels) carrying stimulus-specific information, but with a larger degree of stochasticity, produces more realistic decoding patterns than having fewer dimensions with a lower degree of stochasticity. Here, stochasticity referred to the channels' probability of activation.

the same frequency of response), while on the right column it uses a diversity of frequencies. On the top row, we have a uniform latency of response (i.e. all channels have the same latency of response), whereas on the bottom row we have a diversity of latencies of response. See *Figure 5—figure supplement 1* for a similar result for phase resetting followed by frequency entrainment.

As mentioned, real data normally yield a relatively tight band of high decoding accuracy along the diagonal, often accompanied from contiguous parallel bands of below-baseline accuracy. Critically, the fact that we do not typically observe a chequer pattern means that the trajectory of phase across channels does not repeat itself periodically. If it did, it would show patterns as in the top-left panel, where the uniformity of frequencies and latencies gives rise to an unrealistically regular pattern—such that a decoder trained at a certain time point will become equally accurate again after one cycle at the entrained frequency. Having a diversity of latencies but not of frequencies produces another regular pattern consisting of alternating, parallel bands of higher/lower than baseline accuracy. This, shown in the bottom-left panel, is not what we see in real data either. Having a diversity of frequencies but not of latencies gets us closer to a realistic pattern, as we see in the top-right panel. Finally, having diversity of both frequencies and latencies produces the most realistic pattern, as we can see by comparing to the examples in *Figure 2*, and many others throughout the literature (*King and Dehaene, 2014*). Similar conclusions can be drawn from *Figure 5—figure supplement 1* for phase resetting plus frequency entrainment.

In summary, these results show that it is not only important for the stimulus-relevant subspace of activity to be spatially high dimensional, but also temporally high dimensional.

## A slower additive component is coupled with the oscillatory response

In real data, we normally see a broadening of decoding accuracy in the TGM as time progresses throughout the trial (see *Figure 2*). This is often interpreted as neural representations becoming more stable at latter stages of the trial, which is putatively linked to memory encoding and higher cognitive processes. In *Figure 6*, I show that this effect can be reproduced on synthetic data through the addition of a slowly progressing, non-oscillatory response.

Specifically, I set up a response function such that, after stimulus presentation, the non-oscillatory additive response ramps up to a stimulus-specific target value in about 100 s, and then slowly decays to finally vanish at around 800 ms. Also, I modulate the strength of the oscillatory response using values of $\bar{z}^k$ (see Methods) that differ between the two stimuli by a magnitude that ranges from 0.0 (no difference) to 1.0 (for reference, the examples in the previous figures had a difference of 0.5). As

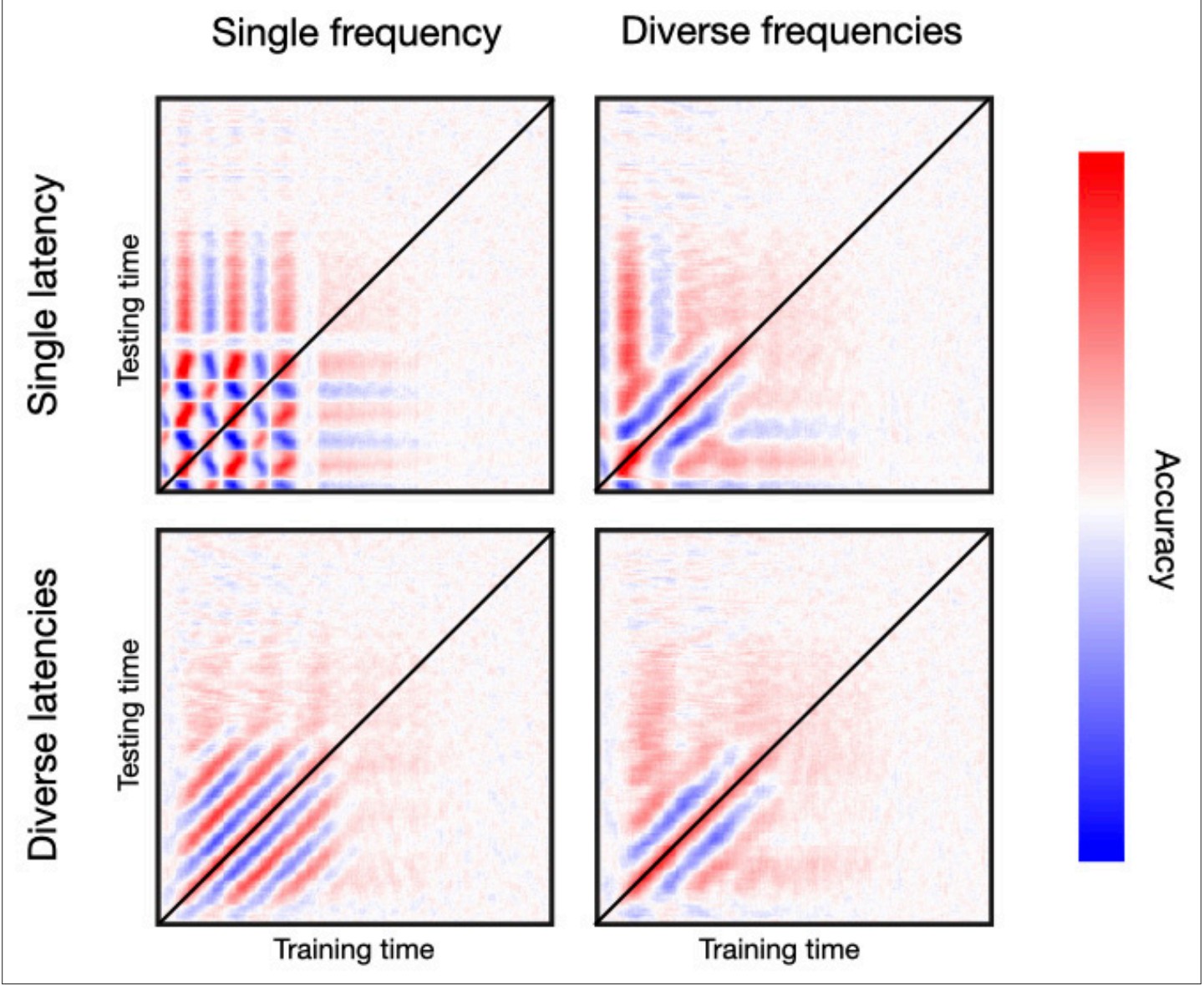

**Figure 5.** Oscillatory responses to stimulation occur in a diversity of frequencies and latencies across channels. Top row, single latency of response; bottom row, diverse latencies across channels. Left column, single frequency; right column, diverse frequencies across channels. Bottom-right is the most realistic temporal generalisation matrix (TGM).

The online version of this article includes the following figure supplement(s) for figure 5:

**Figure supplement 1.** Phase reset plus frequency entrainment effect.

seen in *Figure 6A*, the strength of decoding accuracy grows as the difference in the slow response increases.

Another feature commonly seen on real data are the vertical and horizontal bars of high accuracy stemming from the time point of maximum accuracy (see *Figure 2A*), which is sometimes interpreted as evidence of stimulus representation recurrence in the brain. I show in *Figure 6B* that this feature emerges from a phase coupling between the oscillatory component and the slower component (with respect to the stimuli). For example, following the notation established in Methods, an in-phase relationship means that the sign of $\left(\overline{\varphi^1} - \overline{\varphi^2}\right)$ at $t_{\max}$ is the same than the sign of $\left(\overline{z^1} - \overline{z^2}\right)$. Note that the sign of $\left(\overline{z^1} - \overline{z^2}\right)$ will likely be maintained for most of the trial since this component is slow, therefore creating the effect in the TGM. That was the case for all panels of *Figure 6A*. For

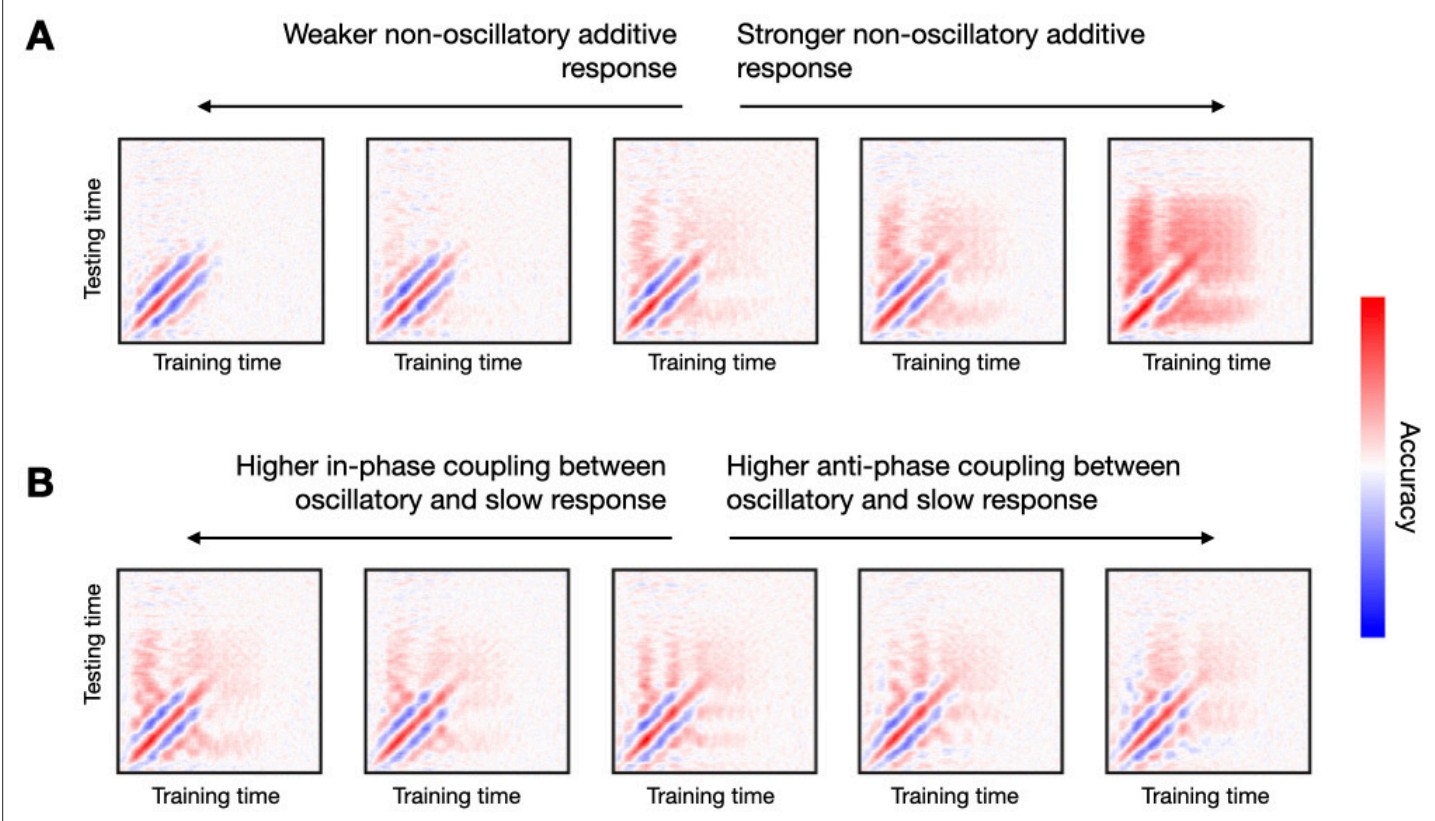

**Figure 6.** An additive non-oscillatory response is needed to produce realistic temporal generalisation matrices (TGMs) with a broadening of decoding accuracy at later stages of the trial. (**A**) By increasing the strength of the non-oscillatory response, the broadening of accuracy becomes more prominent. (**B**) Changing the nature of the phasic relationship between the slower and the faster (oscillatory component) greatly influences the TGM, from having all channels in-phase (left) towards having all channels anti-phase (right).

*Figure 6B*, while keeping $\left(\overline{z^1} - \overline{z^2}\right)$ = 0.5 (as in the middle panel of *Figure 6A*), I varied the phase consistency between the oscillatory and the non-oscillatory components. In the leftmost panel, the oscillatory and the non-oscillatory components are in-phase for all channels, while in the rightmost panel, they are anti-phase for all channels; in between, 25%, 50%, and 75% of the channels are in-phase (and 75%, 50%, and 25% are anti-phase, respectively). As observed, the type of phase consistency between the oscillatory and the non-oscillatory component has a strong impact on the TGM. In particular, the in-phase relation bears the most consistent patterns with the considered real data.

## Amplitude increases modulate the size of the effect

Are modulations in the amplitude of the stimulus-specific oscillation necessary for the effects we observe in real data? They are not, but they can enhance the already existing patterns.

I generated data sets with an additive oscillatory component effect. In each of them, I applied a different amount of amplitude enhancement, from no increase to a fivefold increase. These did not entail any effect on the signals' phase. I considered two characteristic features of the TGM: (1) the diagonal, and (2) a vertical slice stemming from the time point of maximum accuracy. *Figure 7* shows the results. As observed, the main features are present in all runs regardless of the size of the amplitude effect, although they were more prominent for higher amplitude modulations. An amplitude modulation on a phase-resetting effect causes a similar effect (not shown). Note that an amplitude effect without a phase effect would not result in any phase locking across trials, and therefore could not lead to any significant decoding accuracy.

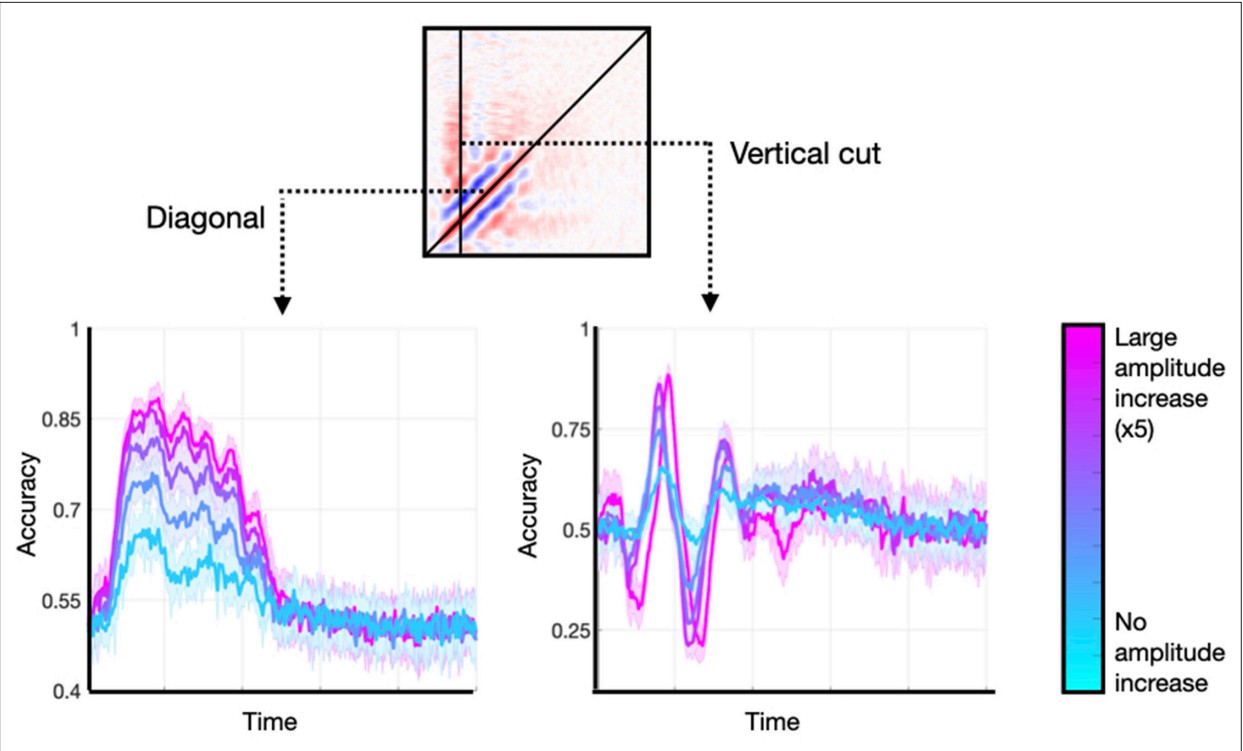

**Figure 7.** Enhancing the amplitude of the stimulus-relevant oscillation is not strictly necessary to produce realistic temporal generalisation matrices (TGMs), but it can enlarge the effects. Two features of the TGM are highlighted: the diagonal, and a vertical slice at the time point of maximum accuracy.

### Fitting an empirical TGM

The previous analyses were descriptive in the sense that they did not quantify how much the generated TGMs resembled a specific empirical TGM. This was deliberate, because empirical TGMs vary across subjects and experiments, and I aimed at characterising them as generally as possible by looking at some characteristic features in broad terms. For example, while TGMs typically have a strong diagonal and horizontal/vertical bars of high accuracy, questions such as when these effects emerge and for how long are highly dependent on the experimental paradigm. For the same reason, I did not optimise the model hyperparameters, limiting myself to observing the behaviour of the model across some characteristic configurations. But, often, one's interests are more specific. Then, it would be interesting to optimise some key hyperparameters of the model to maximise the correlation with a particular empirical TGM.

To illustrate how a data set could be more explicitly considered, I took an empirical TGM computed from the visual experiment in *Cichy et al., 2016*. Specifically, this is a cross-average TGM (over 10 subjects) obtained from decoding animate versus inanimate stimuli. Using an additive oscillatory response, and fixing the rest of the hyperparameters to a sensible configuration (the one that produced the middle panel in *Figure 6A*), I varied, within a reasonable range, two parameters of the response function that control the temporal shape of the effect: the rise slope $\delta_2$ and the fall slope $\delta_3$; see Methods. From each pair of parameters, I generated 10 data sets and computed 10 TGMs; I then correlated the average of these with the empirical TGM from real data. *Figure 8* shows a heatmap of the resulting correlations. For a specific configuration ($\delta_2 = 0.06s, \delta_3 = 0.09s$), the correlation peaks at $r = 0.7$.

### Discussion

Brain's responses to even the simplest stimuli are characteristically variable. This is not surprising, given the brain's plasticity, and the fact that its endogenous activity is ever-changing. It also speaks to the brain's degeneracy (*Edelman and Gally, 2001*)—that is that there are many possible neural trajectories that can achieve a single purpose. How this variability translates into behaviour and experience

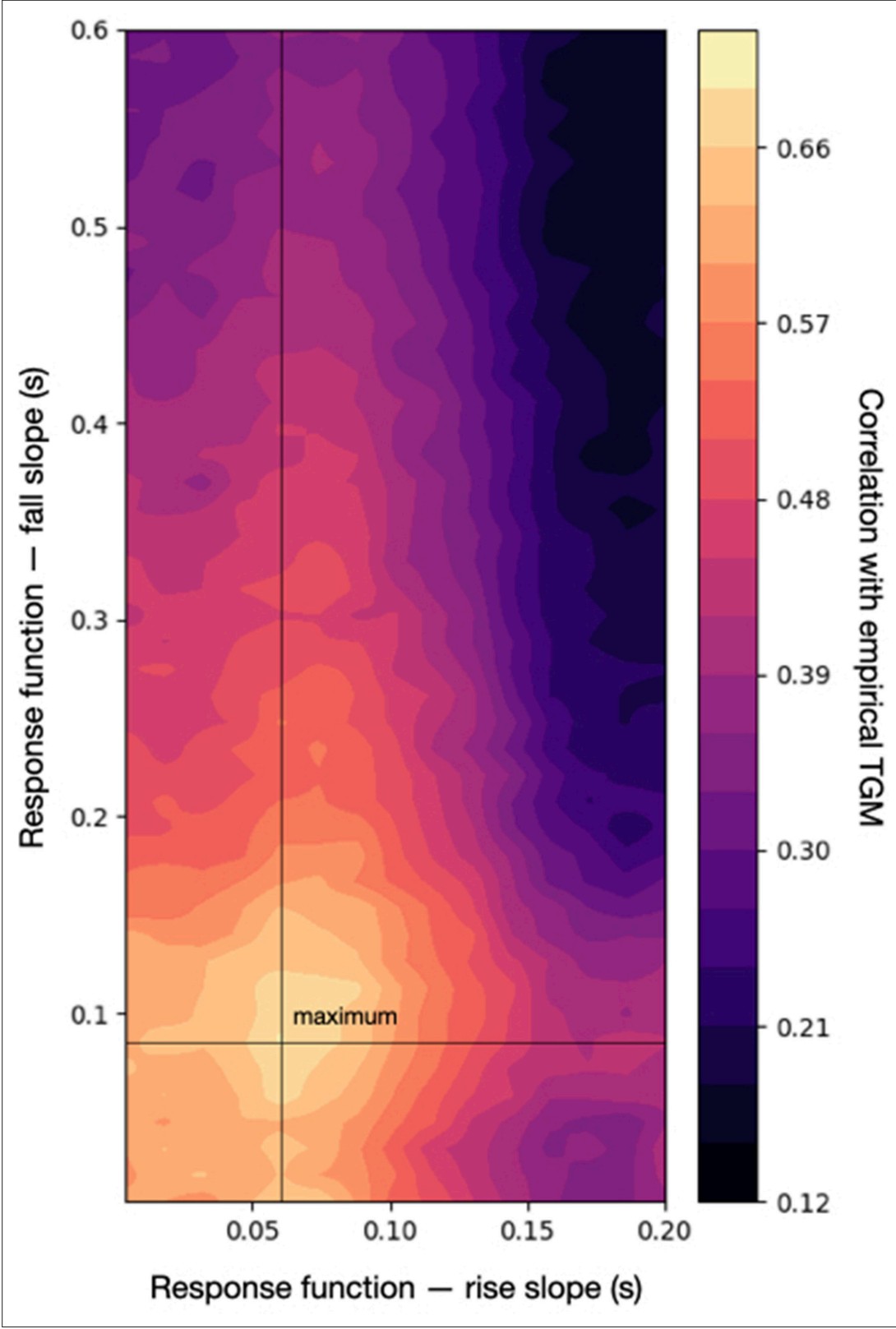

**Figure 8.** Correlation between an empirical temporal generalisation matrix (TGM) and the sampled TGM (averaged across 10 runs of the simulation) across a range of configurations of the response function, which defines the temporal dynamics of the effect. The maximum correlation with the empirical TGM is slightly over 0.7.

is however an open question. Anyhow, it seems reasonable that, at some level, brain responses must keep some invariant aspects so that our perceptual experiences remain stable. Here, I investigated, using a novel generative model, the most stable aspects of brain responses to stimuli as seen through the lens of decoding accuracy, which is, by definition, based on averaging. Previous work has analysed the nature of brain responses to perceptual stimulation using average evoked responses (*Sauseng et al., 2007*), arguing either for a predominant role of additive responses (*Shah et al., 2004*; *Mazaheri and Jensen, 2006*) or phase resetting (*Makeig et al., 2002*). These studies looked at the average response to a given stimulus, but did not investigate what aspects of the data carry information about the identity of the stimulus; this is the focus of this paper and the goal of genephys, the proposed model.

Genephys has different available types of effect, including phase resets, additive damped oscillations, amplitude modulations, and non-oscillatory responses. All of these elements, which may relate to distinct neurobiological mechanisms, are configurable and can be combined to generate a plethora of TGMs that, in turn, can be contrasted to specific empirical TGMs. This way, we can gain insight on what mechanisms might be at play in a given task.

The demonstrations here are not meant to be tailored to a specific data set, and are, for the most part, intentionally qualitative. TGMs do vary across experiments and subjects; and the hyperparameters of the model can be explicitly optimised to specific scientific questions, data sets, and even individuals. In order to explore the space of configurations effectively, an automatic optimisation of the hyperparameter space using, for instance, Bayesian optimisation (*Lorenz et al., 2017*) could be advantageous. This may lead to the identification of very specific (spatial, spectral, and temporal) features in the data that may be neurobiologically interpreted.

Importantly, the list of effects that I have explored here is not exhaustive. For example, I have considered additive oscillatory responses in the form of sinusoidal waves. Another possibility could be to have additive oscillatory responses that are non-linear, that is with a tendency to spend more time in certain phases (e.g. having wider peaks than throughs); in this case, even in the absence of phase locking between trials, we could potentially have a significant evoked response due to phase asymmetry (*Nikulin et al., 2007*). For simplicity, also, I have considered independent sources of activity that exhibit correlations only through the induced effect. In practice, brain areas are in constant communication even in the absence of stimulation. Alternatives where sources are modelled as coupled oscillators (*Breakspear et al., 2010*; *Cabral et al., 2014*), or where there is correlated noise, are also possible. Also for simplicity, I have considered that every channel has one endogenous fundamental frequency only; in practice, multiple ongoing frequencies coexist and interact, potentially affecting the TGM if they are not completely averaged out across trials. Finally, the inherent stochasticity of the stimulus-specific space of activity can take various forms; here, I have explored a probabilistic activation of the channels, but others, such as noise in the phase distribution, are also possible and can also be simulated using *genephys*.

Also importantly, I have shown that standard decoding analysis can differentiate between these explanations only to some extent. For example, the effects induced by phase resetting and the use of additive oscillatory components are not enormously different in terms of the resulting TGMs. In future work, alternatives to standard decoding analysis and TGMs might be used to disentangle these sources of variation (*Vidaurre et al., 2019*).

Overall, the results obtained from applying *genephys* suggest that the stable aspects of brain activity regarding stimulus processing comprise phasic modulations of an oscillatory component coupled with a slower component, with an important role played by the nature of this coupling. The subspace of brain activity induced by the effect is high dimensional in both the spatial domain (it must span many channels) and the frequency domain (it must involve a great diversity of frequencies and exhibit a diversity of latencies). This effect may be accompanied by an amplitude modulation. Above and beyond these average patterns, the stimulus-specific subspace of brain responses remains highly stochastic at the trial level.

## Code accessibility

The model is available as a Python package in PyPI and Github.

## Acknowledgements

DV is supported by a Novo Nordisk Foundation Emerging Investigator Fellowship (NNF19OC-0054895) and an ERC Starting Grant (ERC-StG-2019-850404). I also thank the Wellcome Trust for support (106183/Z/14/Z, 215573/Z/19/Z).

# Additional information

## Funding

| Funder | Grant reference number | Author |
|---|---|---|
| Novo Nordisk Fonden | NNF19OC-0054895 | Diego Vidaurre |
| Horizon 2020 Framework Programme | ERC-StG-2019-850404 | Diego Vidaurre |
| Wellcome Trust | 215573/Z/19/Z | Diego Vidaurre |
| Wellcome Trust | 106183/Z/14/Z | Diego Vidaurre |

The funders had no role in study design, data collection, and interpretation, or the decision to submit the work for publication. For the purpose of Open Access, the authors have applied a CC BY public copyright license to any Author Accepted Manuscript version arising from this submission.

## Author contributions
Diego Vidaurre, Conceptualization, Resources, Data curation, Software, Formal analysis, Funding acquisition, Validation, Investigation, Visualization, Methodology, Writing - original draft, Project administration, Writing - review and editing

## Author ORCIDs
Diego Vidaurre ⓘD http://orcid.org/0000-0002-9650-2229

## Ethics
I used existing data, collected following standard ethical protocols.

Reviewer #1 (Public Review): https://doi.org/10.7554/eLife.87729.3.sa1
Reviewer #2 (Public Review): https://doi.org/10.7554/eLife.87729.3.sa2
Author Response https://doi.org/10.7554/eLife.87729.3.sa3

# Additional files

## Supplementary files
• MDAR checklist

## Data availability
The simulated data can be regenerated using the code included as an example in the genephys Github repository at http://github.com/vidaurre/genephys (copy archived at *Vidaurre, 2024*). The real visual paradigm is openly available from the original publication *Cichy et al., 2016* at: http://userpage.fu-berlin.de/rmcichy/fusion_project_page/main.html.

The following previously published datasets were used:

| Author(s) | Year | Dataset title | Dataset URL | Database and Identifier |
|---|---|---|---|---|
| Cichy RM, Pantazis D, Oliva D | 2016 | Spatio-temporal dynamics of information flow in ventral & dorsal visual cortex - 118 images | http://userpage.fu-berlin.de/rmcichy/fusion_project_page/main.html | Userpage, rmcichy/fusion_project_page/main.html |

*Continued on next page*

*Continued*

| Author(s) | Year | Dataset title | Dataset URL | Database and Identifier |
|---|---|---|---|---|
| Cichy RM, Pantazis D, Oliva D | 2018 | Index of /MEG2_MEG_Epoched_Raw_Data | http://wednesday.csail.mit.edu/MEG2_MEG_Epoched_Raw_Data/ | MIT CSAIL, MEG2_MEG_Epoched_Raw_Data/ |

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
