## [Editor Report · eLife assessment]

This study presents a **valuable** finding on developing a state-of-the-art generative model of brain electrophysiological signals to explain temporal decoding matrices widely used in cognitive neuroscience. The evidence supporting the authors' claims is **convincing**. The results will be strengthened by providing more clear mappings between neurobiological mechanisms and signal generators in the model. The work will be of interest to cognitive neuroscientists using electrophysiological recordings.

---

## [Referee Report · Reviewer #1 (Public Review)]

With genephys, the author provides a generative model of brain responses to stimulation. This generative model allows to mimic specific parameters of a brain response at the sensor level, to test the impact of those parameters on critical analytic methods utilized on real M/EEG data. Specifically, they compare the decoding output for differently set parameters to the decoding pattern observed in a classical passive viewing study in terms of the resulting temporal generalization matrix (TGM). They identify that the correspondence between the mimicked and the experimental TGM to depend on an oscillatory component that spans multiple channels, frequencies, and latencies of response; and an additive, slower response with a specific (cross-frequency) relation to the phase of the oscillatory, faster component.

A strength of the article is that it considers the complexity of neural data that contribute to the findings obtained in stimulation experiments. An additional strength is the provision of a Python package that allows scientists to explore the potential contribution of different aspects of neural signals to obtained experimental data and thereby to potentially test their theoretical assumptions critical parameters that contribute to their experimental data.

A weakness of the paper is that the power of the model is illustrated for only one specific set of parameters, added in a stepwise manner and the comparison to on specific empirical TGM, assumed to be prototypical; And that this comparison remains descriptive. (That is could a different selection of parameters lead to similar results and is there TGM data which matches these settings less well.) It further remained unclear to me, which implications may be drawn from the generative model, following from the capacities to mimic this specific TGM (i) for more complex cases, such as the comparison between experimental conditions, and (ii) about the complex nature of neural processes involved.

Towards this end I would appreciate (i) a more profound explanation of the conclusions that can be drawn from this specific showcase, including potential limitations, as well as wider considerations of how scientists may empower the generative model to (ii) understand their experimental data better and (iii) which added value the model may have in understanding the nature of underlaying brain mechanism (rather than a mere technical characterization of sensor data).

---

## [Referee Report · Reviewer #2 (Public Review)]

This paper introduces a new model that aims to explain the generators of temporal decoding matrices (TGMs) in terms of underlying signal properties. This is important because TGMs are regularly used to investigate neural mechanisms underlying cognitive processes, but their interpretation in terms of underlying signals often remains unclear. Furthermore, neural signals are often variant over different instances of stimulation despite behaviour being relatively stable. The author aims to tackle these concerns by developing a generative model of electrophysiological data and then showing how different parameterizations can explain different features of TGMs. The developed technique is able to capture empirical observations in terms of fundamental signal properties. Specifically, the model shows that complexity is necessary in terms of spatial configuration, frequencies and latencies to obtain a TGM that is comparable to empirical data.

The major strength of the paper is that the novel technique has the potential to further our understanding of the generators of electrophysiological signals which are an important way to understand brain function. The paper clearly outlines how the method can be used to capture empirical data. Furthermore, the used techniques are state-of-the-art and the developed model is publicly shared in open source code.

On the other hand, there is no unambiguous mapping between neurobiological mechanisms and different signal generators, making it hard to draw firm conclusions about neural underpinnings based on this analysis.

---

## [Author Response]

The following is the authors’ response to the original reviews.

**Reviewer #1:**
A weakness of the paper is that the power of the model is illustrated for only one specific set of parameters, added in a stepwise manner and the comparison to one specific empirical TGM, assumed to be prototypical; And that this comparison remains descriptive. (That is could a different selection of parameters lead to similar results and is there TGM data which matches these settings less well.)

The fact that the comparisons in the paper are descriptive is a central point of criticism from both reviewers. As mentioned in my preliminary response, I intentionally did not optimise the model to a specific TGM or show an explicit metric of fitness. As I now explicitly mention in the new experimental section of the paper:

“The previous analyses were descriptive in the sense that they did not quantify how much the generated TGMs resembled a specific empirical TGM. This was deliberate, because empirical TGMs vary across subjects and experiments, and I aimed at characterising them as generally as possible by looking at some characteristic features in broad terms. For example, while TGMs typically have a strong diagonal and horizontal/vertical bars of high accuracy, questions such as when these effects emerge and for how long are highly dependent on the experimental paradigm. For the same reason, I did not optimise the model hyperparameters, limiting myself to observing the behaviour of the model across some characteristic configurations”

And, in the Discussion:

“The demonstrations here are not meant to be tailored to a specific data set, and are, for the most part, intentionally qualitative. TGMs do vary across experiments and subjects; and the hyperparameters of the model can be explicitly optimised to specific scientific questions, data sets, and even individuals. In order to explore the space of configurations effectively, an automatic optimisation of the hyperparameter space using, for instance, Bayesian optimisation (Lorenz, et al., 2017) could be advantageous. This may lead to the identification of very specific (spatial, spectral and temporal) features in the data that may be neurobiologically interpreted.”

Nonetheless, it is possible to fit the model to a specific TGMs by using a explicit metric of fitness. For illustration, this is what I did in the new experimental section Fitting and empirical TGM, where I used correlation with an empirical TGM to optimise two temporal parameters: the rise slope and the fall slope. As can be seen in the Figure 8, the correlation with the empirical TGM was as high as 0.7, even though I did not fit the other parameters of the model. As mentioned in the paragraph above, more sophisticated techniques such as Bayesian optimisation might be necessary for a more exhaustive exploration, but this would be beyond the scope of the current paper.

I would also like to point out that fitting the parameters in a step-wise manner was a necessity for interpretation. I suggest to think of the way we use F-tests in regression analyses as a comparison: if we want to know how important a feature is, we compare the model with and without this feature and see how much we loss.

It further remained unclear to me, which implications may be drawn from the generative model, following from the capacities to mimic this specific TGM (i) for more complex cases, such as the comparison between experimental conditions, and (ii) about the complex nature of neural processes involved.

Following on the previous points, the object of this paper (besides presenting the model and the associated toolbox) was not to mimic a specific TGM, but to characterise the main features that we generally see across studies in the field. To clarify this, I have added Figure 2 (previously a Supplemental Information figure), and added the following to the Results section:

“Figure 2 shows a TGM for an example subject, where some archetypal characteristics are highlighted. In the experiments below, specifically, I focus on the strong narrow diagonal at the beginning of the trial, the broadening of accuracy later in the trial, and the vertical/horizontal bars of higher-than-chance accuracy. Importantly, this specific example in Figure 2 is only meant as a reference, and therefore I did not optimise the model hyperparameters to this TGM (except in the last subsection), or showed any quantitative metric of similarity.”

I mention the possibility of using the model to explore more complex cases in the Introduction, although doing so here would be out of scope:

“Other experimental paradigms, including motor tasks and decision making, can be investigated with genephys”

Towards this end, I would appreciate (i) a more profound explanation of the conclusions that can be drawn from this specific showcase, including potential limitations, as well as wider considerations of how scientists may empower the generative model to (ii) understand their experimental data better and (iii) which added value the model may have in understanding the nature of underlying brain mechanism (rather than a mere technical characterization of sensor data).

To better illustrate how to use genephys to explore a specific data set, I have added a section (Fitting an empirical TGM) where I show how to fit specific hyperparameters to an empirical TGM in a simple manner.

In the Introduction, I briefly mentioned:

“This (not exhaustive) list of effects was considered given previous literature (Shah, et al., 2004; Mazaheri & Jensen, 2006; Makeig, et al., 2002; Vidaurre, et al., 2021), and each effect may be underpinned by distinct neural mechanisms. For example, it is not completely clear the extent to which stimulus processing is sustained by oscillations, and disentangling these effects can help resolving this question”

In the Discussion, I have further commented:

“Genephys has different available types of effect, including phase resets, additive damped oscillations, amplitude modulations, and non-oscillatory responses. All of these elements, which may relate to distinct neurobiological mechanisms, are configurable and can be combined to generate a plethora of TGMs that, in turn, can be contrasted to specific empirical TGMs. This way, we can gain insight on what mechanisms might be at play in a given task.

The demonstrations here are not meant to be tailored to a specific data set, and are, for the most part, intentionally qualitative. TGMs do vary across experiments and subjects; and the hyperparameters of the model can be explicitly optimised to specific scientific questions, data sets, and even individuals. In order to explore the space of configurations effectively, an automatic optimisation of the hyperparameter space using, for instance, Bayesian optimisation (Lorenz, et al., 2017) could be advantageous. This may lead to the identification of very specific (spatial, spectral and temporal) features in the data that may be neurobiologically interpreted. “

On p. 15 "Having a diversity of frequencies but not of latencies produces another regular pattern consisting of alternating, parallel bands of higher/lower than baseline accuracy. This, shown in the bottom left panel, is not what we see in real data either. Having a diversity of latencies but not of frequencies gets us closer to a realistic pattern, as we see in the top right panel." The terms frequency and latency seem to be confused.

The Reviewer is right. I have corrected this now. Thank you.

**Reviewer #2:**
The results of comparisons between simulations and real data are not always clear for an inexperienced reader. For example, the comparisons are qualitative rather than quantitative, making it hard to draw firm conclusions. Relatedly, it is unclear whether the chosen parameterizations are the only/best ones to generate the observed patterns or whether others are possible. In the case of the latter, it is unclear what we can actually conclude about underlying signal generators. It would have been different if the model was directly fitted to empirical data, maybe of different cognitive conditions. Finally, the neurobiological interpretation of different signal properties is not discussed. Therefore, taken together, in its currently presented form, it is unclear how this method could be used exactly to further our understanding of the brain.

This critique coincides with that of Reviewer 1. In the current version, I made more clear the fact that I am not fitting a specific empirical TGM and why, and that, instead, I am referring to general features that appear broadly throughout the literature. See more detailed changes below.

Regarding whether the chosen parameterizations are the only/best ones to generate the observed patterns, the Discussion reflects this limitation:

“Also importantly, I have shown that standard decoding analysis can differentiate between these explanations only to some extent. For example, the effects induced by phase-resetting and the use of additive oscillatory components are not enormously different in terms of the resulting TGMs. In future work, alternatives to standard decoding analysis and TGMs might be used to disentangle these sources of variation (Vidaurre, et al., 2019). ”

And

“Importantly, the list of effects that I have explored here is not exhaustive …”

Of course, since the list of signal features I have explored is not exhaustive, it cannot be claimed without a doubt that these features are the ones generating the properties we observe in real TGMs. The model, however, is a step forward in that direction, as it provides us with a tool to at least rule out some causes.

Firstly, it was not entirely clear to me from the introduction what gap exactly the model is supposed to fill: is it about variance in neural responses in general, about which signal properties are responsible for decoding, or about capturing stability of signals? It seems like it does all of these, but this needs to be made clearer in the introduction. It would be helpful to emphasize exactly what insights the model can provide that are unable to be obtained with the current methods.

I have now made this explicit in in the Introduction, as suggested:

“To gain insight into what aspects of the signal underpin decoding accuracy, and therefore the most stable aspects of stimulus processing, I introduce a generative model”

To help illustrating what insights the model can provide, I have added the following sentence as an example:

“For example, it is not completely clear the extent to which stimulus processing is sustained by oscillations, and disentangling these effects can help resolving this question.”

Furthermore, I was unclear on why these specific properties were chosen (lines 71 to 78). Is there evidence from neuroscience to suggest that these signal properties are especially important for neural processing? Or, if the logic has more to do with signal processing, why are these specific properties the most important to include?

To clarify this the text now reads:

“In the model, when a channel responds, it can do it in different ways: (i) by phase-resetting the ongoing oscillation to a given target phase and then entraining to a given frequency, (ii) by an additive oscillatory response independent of the ongoing oscillation, (iii) by modulating the amplitude of the stimulus-relevant oscillations, or (iv) by an additive non-oscillatory (slower) response. This (not exhaustive) list of effects was considered given previous literature (Shah, et al., 2004; Mazaheri & Jensen, 2006; Makeig, et al., 2002; Vidaurre, et al., 2021), and each effect may be underpinned by distinct neural mechanisms”

The general narrative and focus of the paper could also be improved. It might help to start off with an outline of what the goal is at the start of the paper and then explicitly discuss how each of the steps works toward that goal. For example, I got the idea that the goal was to capture specific properties of an empirical TGM. If this was the case, the empirical TGM could be placed in the main body of the text as a reference picture for all simulated TGMs. For each simulation step, it could be emphasized more clearly exactly which features of the TGM is captured and what that means for interpreting these features in real data.

Thank you. To clarify the purpose of the paper better, I have brought Figure 2 to the front (before a Supplementary Figure), and in the first part of Results I have now added:

“Figure 2 shows a TGM for an example subject, where some archetypal characteristics are highlighted. In the experiments below, specifically, I focus on the strong narrow diagonal at the beginning of the trial, the broadening of accuracy later in the trial, and the vertical/horizontal bars of higher-than-chance accuracy. Importantly, this specific example in Figure 2 is only meant as a reference, and therefore I did not optimise the model hyperparameters to this TGM (except in the last subsection), or showed any quantitative metric of similarity. ”

I have enunciated the goals more clearly in the Introduction:

“To gain insight into what aspects of the signal underpin decoding accuracy, and therefore the most stable aspects of stimulus processing, …”

Relatedly, it would be good to connect the various signal properties to possible neurobiological mechanisms. I appreciate that the author tries to remain neutral on this in the introduction, but I think it would greatly increase the implications of the analysis if it is made clearer how it could eventually help us understand neural processes.

The Reviewer is right in pointing out that I preferred to remain neutral on this. While I have still kept that tone of neutrality throughout the paper, I have now included the following sentence as an example of a neurobiological question that could be investigated with the model:

“For example, it is not completely clear the extent to which stimulus processing is sustained by oscillations, and disentangling these effects can help resolving this question.”

And, more generally,

“Genephys has different available types of effect, including phase resets, additive damped oscillations, amplitude modulations, and non-oscillatory responses. All of these elements, which may relate to distinct neurobiological mechanisms, are configurable and can be combined to generate a plethora of TGMs that, in turn, can be contrasted to specific empirical TGMs. This way, we can gain insight on what mechanisms might be at play in a given task. ”

Line 57: this sentence is very long, making it hard to follow, could you break up into smaller parts?

Thank you. The sentence is fragmented now.

Please replace angular frequencies with frequencies in Hertz for clarity.

Here I have preferred to stick to angular frequencies because it is more general than if I talk about Hertz, because that would entail having a specific sampling frequency. I think doing so would create confusion precisely of the sorts that I am trying to clarify in this revision: that is, that these results are not specific of one TGM but reflect general features that we see broadly in the literature.

There are quite some types throughout the paper, please recheck

Thank you. I have revised and have made my best to clear them out.